# Liposome-Encapsulated Antibiotics for the Therapy of Mycobacterial Infections [note 1]

**DOI:** 10.3390/antibiotics14070728

**Published:** 2025-07-20

**Authors:** Metin Yıldırım, Nejat Düzgüneş

**Affiliations:** 1Department of Biochemistry, Faculty of Pharmacy, Harran University, Şanlıurfa 63000, Türkiye; metinyildirim4@gmail.com; 2Department of Biomedical Sciences, Arthur A. Dugoni School of Dentistry, University of the Pacific, 155 Fifth Street, Francisco, CA 94103, USA

**Keywords:** liposomes, tuberculosis, antibiotics, antimycobacterial agents, bacterial infections, tuberculosis vaccines

## Abstract

About a quarter of the world’s population is infected with *Mycobacterium tuberculosis*. Growing antibiotic resistance by this microorganism is a major problem in the therapy of the disease. *M. avium-M. intracellulare* that emerged as a major opportunistic infection of HIV/AIDS continues to afflict immunocompromised individuals. We describe the use of liposome-encapsulated antibiotics in the experimental and clinical therapy of mycobacterial infections, as well as recent experimental liposomal vaccines against tuberculosis. Liposome-mediated intravenous or inhalational delivery of antibiotics enhances the antibacterial effects of the drugs, particularly for infections of resident macrophages, where the liposomes are passively targeted. Despite experimental successes of liposomal antibiotics in the treatment of mycobacterial and other bacterial infections, applications of this method to the clinic have been lagging. This review underscores the significance of liposomes in the treatment of mycobacterial infections, encompassing their synthesis methods, limitations, and both preclinical and clinical studies, providing guidance for the development of future therapeutic approaches and innovative antimicrobial strategies.

## 1. Mycobacteria

Mycobacteria are aerobic bacteria with a diameter of approximately 0.2–0.6 µm and length between 1 and 10 µm. Owing to the long-chain lipids, termed mycolic acids, in the bacterial membrane, the surface of the microorganism presents a hydrophobic interface, which resists the entry of many disinfectants as well as the Gram and Giemsa stains. A major characteristic of mycobacteria is that after incubation with the Ziehl–Neelsen stain, they are resistant to the decolorizing effect of an ethanol-hydrochloric acid solution (which can remove the stain from many other bacteria) and are thus termed acid-fast bacteria. Nevertheless, under some conditions, for example, after treatment with the mycolic acid synthesis inhibitor, isoniazid, or when it is in a metabolically inactive state in the host, *M. tuberculosis* may become acid-fast-negative [1]. Mycobacteria grow very slowly. The doubling time for *Mycobacterium tuberculosis* may be as long as 1676 h in the chronic infection phase in infected mice, and about 25 h during acute infection [2].

During the Industrial Revolution in the 18th and 19th centuries in Europe, tuberculosis became an epidemic, exacerbated by the crowding of urbanization and economic depression [3]. One quarter of the population of Europe in the 19th Century is thought to have died of “consumption,” as tuberculosis was called at the time. The victims included Goethe, Rousseau, Thoreau, Keats, Paganini, and Chopin. It is estimated that even in the 21st Century, one in four people in the world are infected with *M. tuberculosis*. According to the World Health Organization, the bacterium infected 10.8 million people in 2023 and killed about 1.25 million people. Among these, 161,000 people were also infected with HIV-1.

It is estimated that between 5 and 10% of individuals infected with the mycobacterium will eventually develop tuberculosis. The incidence is highest in Southeast Asia (45%), Africa (24%), and the Western Pacific (17%). Tuberculosis is prevalent in areas with poverty, malnutrition and poor housing. Mycobacteria have a complex cell wall, composed of the peptidoglycan layer linked with arabinose-galactose-mycolic acid [3]. Long chain mycolic acids, in the range of 70 to 90 carbon atoms, are the major lipids in the mycobacterial cell wall. Free lipids are located on the outer layers, including waxes, mycosides, which are complex saturated glycolipids, and 6,6′-dimycolate of trehalose, known as cord factor, one of the virulence factors of mycobacteria.

The tubercle bacillus is transmitted by the inhalation of infectious aerosols, generally involving person-to-person contact in close quarters, and occasionally by ingestion and skin trauma. Large aerosol particles are usually trapped by the lung mucosal surfaces and removed by the mucociliary escalator. Smaller aerosol particles with 1–3 mycobacteria can reach the alveoli and are phagocytosed by alveolar macrophages, where they replicate and may destroy their host cell. Infected macrophages may migrate to local lymph nodes, the bloodstream, bone marrow, spleen, kidneys, and the central nervous system [3].

According to American Thoracic Society recommendations, the initial treatment regimen involves isoniazid (and inhibitor of mycolic acid biosynthesis), rifampin (an inhibitor of bacterial RNA synthesis), pyrazinamide (an inhibitor of fatty acid synthase type I), and ethambutol (an inhibitor of arabinogalactan synthesis) for 2 months. This is followed by isoniazid and rifampin therapy for 4–6 months. Noncompliance is known to cause the emergence of multi-drug-resistant (MDR) strains; thus, in many countries, directly observed therapy (DOT) is employed to ensure compliance.

MDR strains that are resistant to at least isoniazid and rifampin have become a worldwide problem [4]. The resistance is attributed to one or more chromosomal mutations. One of these mutations is in a gene for mycolic acid synthesis (*inhA*), and another in a gene for catalase-peroxidase (*katG*), an enzyme required to activate isoniazid within the bacterium [3]. Previous treatment for tuberculosis predisposes the patient to the selection of MDR organisms. In addition to MDR tuberculosis, extensively drug resistant (XDR) *M. tuberculosis* strains have emerged throughout the world [5]. These bacteria are MDR-*M. tuberculosis* strains that are resistant to fluoroqinolones and at least one of the second line drugs, such as kanamycin, capreomycin and amikacin.

*M. avium* and *M. intracellulare* (MAC) are difficult to differentiate physiologically, and cause identical diseases, and are thus grouped together as the *M. avium-M. intracellulare* Complex, abbreviated as MAC [3]. They cause disseminated disease, where tissue macrophages are inundated with these microorganisms, especially in advanced HIV/AIDS patients, and the blood contains large numbers of organisms. *M. abscessus* has been recognized recently as a pathogen that can colonize the lungs of patients with cystic fibrosis, chronic obstructive pulmonary disease, or bronchiectasis, and that can grow in macrophages and free-living amoebae [6]. The mycobacterium has intrinsic and acquired mechanisms of resistance to therapeutics [7]. In addition to conventional drug therapies, lipid-based drug delivery systems—such as polymeric carriers, liposomes, solid lipid nanoparticles, and niosomes—have been increasingly utilized in the treatment of mycobacterial infections in recent years.

Liposomes exhibit high biocompatibility and low toxicity. Furthermore, they are readily phagocytosed by macrophages, making them particularly suitable for targeting intracellular pathogens such as *M. tuberculosis*. One of their key advantages is the ability to encapsulate both hydrophilic and hydrophobic drugs. In addition to intravenous administration, their suitability for inhalational delivery offers a significant therapeutic benefit. Several liposomal formulations have been approved by the FDA and are available as commercial products.

## 2. Early Studies on Liposome-Encapsulated Antibiotics for Tuberculosis Therapy

The first English-language publications on the use of liposome-encapsulated antibiotics appeared in 1982. Vladimirsky and Ladigina [8] treated mice infected with *M. tuberculosis* strain H37Rv by intravenous injection of streptomycin sulfate encapsulated in liposomes composed of lecithin (phosphatidylcholine with various acyl chains). There was a statistically different decrease in mycobacterial counts in the spleen compared to an equivalent concentration of free streptomycin, which translated into prolonged survival in the liposomal antibiotic group, and reduced antibiotic toxicity. In a subsequent study, Ladigina and Vladimirsky [9] showed that the total area under the serum concentration-time curve (“AUC”) of liposomal ^3^H-dihydrostreptomycin was 8.8 times higher in uninfected mice than the free antibiotic, and 5.9 times higher in mice with advanced tuberculosis. The total amounts of liposome-delivered antibiotic in the spleen and liver of infected mice were 9.2 and 7.3 times higher, respectively, than that achieved with the free antibiotic. Orozco et al. [10] reported that mice with severe tuberculosis treated with rifampicin and isoniazid in free and liposome-encapsulated form, together, had a higher survival rate (about 85%) after 30 days, the lowest colony-forming units (CFU) of the bacteria, and less inflammation in the lungs compared to untreated controls. These early experiments were followed by the impressive results obtained by Agarwal et al. [11], who employed rifampin-loaded phosphatidylcholine liposomes to which the macrophage activating tetrapeptide, tuftsin (Thr-Lys-Pro-Arg), was coupled covalently via its C-terminus, using an ethylene diamine spacer. The liposomes delivered twice weekly for 2 weeks were several orders of magnitude more effective than the free drug in lowering the CFU in the lungs, liver, and spleen of mice that had been infected for 13–16 days with the virulent H37Rv strain of *M. tuberculosis*.

These pioneering studies indicated that the passive targeting of intravenously administered liposomes to the reticuloendothelial system because of their particulate and biodegradable nature mediate the delivery of the encapsulated antibiotics to the very sites occupied by the infecting mycobacteria.

## 3. Early Studies on Liposomal Antibiotic Therapy of *Mycobacterium avium-Mycobacterium intracellulare* Complex Infections

The first report on the use of liposome-encapsulated antibiotics against experimental MAC infections in beige mice employed amikacin in phosphatidylglycerol (PG)-phosphatidlycholine (PC)-cholesterol (chol) (1:1:1) liposomes prepared by reverse-phase evaporation, followed by extrusion through polycarbonate membranes [12]. This formulation, delivered intravenously, arrested the growth of MAC in the liver, and reduced the CFU counts by about 1000-fold in the spleen and kidneys, compared with those of both untreated controls and free-drug-treated mice. Using a much higher dose of amikacin, either free or encapsulated in phosphatidylcholine liposomes, Cynamon et al. obtained similar results in the liver and spleen; however, they also found a significant reduction in CFU in the lungs with both free and liposomal amikacin [13]. Relatively low concentrations of amikacin and gentamicin in liposomes reduced significantly the MAC CFU in blood, liver, and spleen [14]. Klemens et al. found that both encapsulated and free gentamicin reduced viable MAC counts in the liver, spleen and lungs compared with no treatment [15]. Encapsulated gentamicin was more effective than the free antibiotic in reducing the viable cell counts in the spleen and liver. Liposomal amikacin was also more effective than the free drug when administered to MAC-infected murine peritoneal macrophages [16].

Intraperitoneal administration of rifampicin (RIF)-incorporating multilamellar liposomes composed of egg PC:dicetylphosphate:chol, and incorporating RIF either in the membrane or in the aqueous interior resulted in a larger reduction in bacterial growth in the lungs and spleen of infected ddY mice than did free RIF. Liposomal delivery resulted in a larger incorporation of RIF into peritoneal macrophages and enhanced killing of intracellular bacteria [17]. Liposome-encapsulated streptomycin, administered intravenously in weekly doses (15 mg/kg) for 4 weeks, reduced the CFU in the liver and spleen of MAC-infected beige mice by an extent similar to that of a 50- to 100-fold higher dose of the free drug [18]. With this injection schedule, the CFU in the liver and spleen were lower by 2.4 and 2.9 log units, respectively, compared to untreated controls, even by the end of 12 weeks, suggesting that the liposomal formulation also increases the residual activity of the drug in these organs. In a parallel study, the effect of free streptomycin at 150 mg/kg given intramuscularly five days a week for 8 weeks was compared with 15 mg/kg of streptomycin in unilamellar liposomes administered intravenously in four injections, with no further treatment up to 8 weeks [19]. The chemotherapeutic efficacy, expressed as the reduction in CFU/unit dose of the antibiotic, was several fold higher for the liposomal drug.

Weekly injections of liposome-encapsulated kanamycin resulted in the reduction in gross pulmonary lesions to a greater extent than the free antibiotic or the drug injected together with empty liposomes [20]. As expected, liposomal delivery of kanamycin caused a greater accumulation and retention of the antibiotic in the liver, spleen and lungs compared to the free drug.

## 4. Encapsulation of Antibiotics in Liposomes and Possible Liposome “Fusion” with Bacteria

Various drug delivery systems, including metallic, polymeric, carbon-based, and dendrimeric carriers, as well as liposomes, have been utilized to overcome the limitations of antibiotics, such as low aqueous solubility, drug degradation, and antibiotic resistance, while simultaneously enhancing their bioavailability and minimizing adverse effects [21]. Liposomes offer advantages such as biocompatibility, capacity for self-assembly, low immunogenicity, passive and active targetability, prolonged half-life of the loaded drug, protection of sensitive molecules, and enhanced bioavailability [22].

Liposomes are generally classified based on their size, with small unilamellar vesicles (SUVs) being <100 nm and large unilamellar vesicles (LUVs) being >100 nm, as well as by the number of lamellae (unilamellar or multilamellar vesicles). In the preparation of liposomal formulations, the composition and charge—neutral, anionic, or cationic—can be tailored according to the therapeutic target [23]. This is particularly important for the interaction of liposomes with bacteria and their uptake by eukaryotic cells [24]. Additionally, liposomal formulations offer significant advantages in the optimal treatment of infections, as they can be administered through various routes, including intravenous [12,19], subcutaneous [25], transdermal [26], oral [27], inhalational [28,29], and nasal delivery [30].

The encapsulation of antibiotics in liposomes can be achieved through two main approaches: active and passive loading. Passive loading techniques include mechanical dispersion, solvent dispersion, and detergent removal methods, whereas active loading techniques involve approaches such as detergent dialysis and microfluidic methods. The latter techniques are used to load drugs into preformed liposomes, ensuring minimal drug loss during the loading process [31].

Hydrophilic antibiotics are incorporated into the aqueous core, whereas hydrophobic antibiotics are naturally embedded within the lipid bilayer. The development of antibiotic-loaded liposomes depends on multiple factors, including the physicochemical properties of the drug, formulation stability, drug leakage, and retention [32]. In general, liposomes are prepared by dissolving lipid components in an organic solvent, forming a lipid film, which is then redispersed in an aqueous medium, followed by sizing and purification. Various techniques have been utilized for liposome synthesis and antibiotic encapsulation, including sonication, thin-film hydration, freeze-thawing, micro-emulsification, solvent injection, reverse-phase evaporation, dehydration–rehydration, hydration in a packed bed of colloidal particles, pH jumping, detergent removal, and extrusion [33,34].

Due to key formulation factors such as drug loading efficiency, scalability, and particle size control, various liposome synthesis methods exhibit specific advantages and limitations, particularly when evaluated for clinical translation and large-scale production. Among the conventional techniques, the thin-film hydration method holds a well-established position, with several FDA-approved liposomal formulations such as Doxil^®^ and AmBisome^®^ developed using this approach. Its historical precedence and regulatory familiarity make it advantageous for clinical progression. However, this technique is often time-consuming, requires stringent control of residual organic solvents, and poses challenges in achieving sterile final products, especially during scale-up.

Methods such as reverse-phase evaporation and dehydration–rehydration are known for their relatively high encapsulation efficiency, particularly for hydrophilic compounds. Nevertheless, the reverse-phase evaporation technique involves extensive use of organic solvents, which may raise toxicity and compatibility concerns, especially for thermolabile or fragile biomolecules. The dehydration–rehydration method, while effective in encapsulating macromolecules, remains largely confined to laboratory-scale studies due to its operational complexity and limited scalability.

From the standpoint of encapsulation performance, reverse-phase evaporation, dehydration–rehydration, and supercritical fluid-based techniques emerge as superior. Among these, the supercritical fluid method offers multiple advantages, including fine control over particle size, reduced or negligible use of toxic organic solvents (particularly when GRAS-class solvents are used), and inherently aseptic processing conditions that alleviate the need for terminal sterilization. Despite its promise as a green and tunable encapsulation technology, the broader adoption of this method is currently limited by high equipment costs and the need for extensive optimization to tailor process parameters for different drug-carrier systems [35,36].

One of the distinguishing factors that set liposomes apart from other drug carriers in antibiotic delivery is the flexibility of their surface modification. The frequently employed polyethylene glycol (PEG)-conjugated lipids can be used to confer liposomes the property of prolonged circulation after intravenous administration [37,38]. These “PEGylated” liposomes can be further functionalized on their surface with antibodies, peptides, proteins, or carbohydrates [39]. This surface modification may enhance drug targeting, thereby improving therapeutic effectiveness [40,41,42].

Bacteria can develop resistance mechanisms against drugs through various pathways, such as enzymatic inactivation of the drug and active efflux pumping, which expels the drug from the cell [43]. Numerous studies have demonstrated that liposome-based antibiotic formulations can be effective in treating antibiotic-resistant bacteria (Figure 1) [44].

Antibiotics encapsulated within liposomes can target both Gram-negative and Gram-positive bacteria, which often produce enzymes that degrade antimicrobial agents. Additionally, for an antibiotic to exert its effect, it must penetrate the bacterial membrane and enter the cell. However, mutations can lead to alterations in outer membrane porins, reducing permeability [45]. In Gram-negative bacteria, the complex structure of the outer membrane further hinders antibiotic penetration, contributing to resistance. Moreover, active efflux pump proteins, which play a crucial role in bacterial physiology, can also contribute to resistance by expelling antibiotics before they reach their target. During the formulation of liposomes, the use of potentially fusogenic phospholipids may help overcome this challenge by facilitating membrane disruption. These phospholipids may facilitate the fusion of liposomes with bacterial membranes, improving drug delivery and increasing the intracellular uptake of antibiotics, thereby enhancing their therapeutic efficacy against resistant bacterial strains [46].

Although one would expect that the peptidoglycan layer of Gram-positive bacteria and the lipopolysaccharide layer of the outer membrane of Gram-negative bacteria might both be inhibitory to the close apposition of liposomal and bacterial membranes, a prerequisite for fusion [47], several studies have provided evidence for membrane fusion. Ma et al. [48] utilized a fluorescent lipid-dilution assay [49,50] to monitor lipid mixing between liposomes (composed of dipalmitoylphosphatidylcholine (DPPC), dioleoylphosphatidylethanolamine (DOPE), dimyristoylphosphatidylglycerol (DMPG), and the fluorescence resonance energy pair, NBD-phosphatidylethanolamine and Rhodamine-phosphatidylethanolamine) and *Pseudomonas aeruginosa*. They found that the presence of DOPE and calcium ions (5 mM), especially with high mol fractions of DMPG greatly stimulated lipid mixing. Utilizing liposomes composed of DPPC:DMPG (18:1), Suchetelli et al. [51] demonstrated the delivery of encapsulated tobramycin into *P. aeruginosa* by electron microscopy, although a prolonged incubation on the order of hours was necessary. The latter experiments were performed in protease peptone 2, which may include calcium. In the lipid mixing assays, however, no calcium was included in the medium.

Given the challenges of antibiotic resistance, liposomal antibiotic formulations have been explored widely as a potential strategy to enhance the efficacy of antibiotics against resistant bacterial strains (Table 1). In a study targeting *Staphylococcus aureus* biofilms, negatively charged liposomes encapsulating levofloxacin and vancomycin were prepared using the dehydration–rehydration technique, enabling in situ antibiotic release within the biofilm [52]. Additionally, nafcillin-loaded PEG-grafted liposomes, prepared via the reverse-phase evaporation method with an average size of 253 nm, exhibited a fourfold reduction in the minimum inhibitory concentration (MIC) against methicillin-susceptible *Staphylococcus aureus* (MSSA) compared to free nafcillin [53].

For the treatment of both Gram-positive and Gram-negative bacterial infections, amoxicillin, a β-lactam antibiotic, was successfully encapsulated into liposomes (~200 nm in size) using the Supercritical Assisted Liposome Formation (SuperLip) method. In this technique, liposomes are prepared by spraying water droplets (containing the antibiotic) into an expanded phase composed of phospholipids, ethanol, and carbon dioxide (CO_2_) under high pressure. During the process, the water droplets are rapidly surrounded by a lipid layer, and upon falling into the water pool located at the bottom of the vessel, liposomes are formed. Using this approach, an encapsulation efficiency of 84% was achieved [54].

In a very different type of application of liposomal antibiotics, Wang et al. [55] used liposome-encapsulated silver-tinidazole to kill colorectal tumor-associated *Fusobacterium nucleatum* in an animal model, thereby generating microbial neoantigens that elicited anti-tumor CD8^+^ T cells. Interestingly, the T cells could recognize both infected and uninfected tumors. Tinidazole was encapsulated in liposomes using a technique called “remote-loading,” originally developed by Barenholz and colleagues for the entrapment of doxorubicin, using an ammonium sulfate gradient that facilitated the generation of a 100-fold higher concentration of the drug inside the liposomes [56]. For tinizadole entrapment, Wang et al. used a silver nitrate gradient. The authors indicated that the liposomes were pH-sensitive, which would facilitate antibiotic release in the low pH-environment of the tumor, but did not disclose their lipid composition.
antibiotics-14-00728-t001_Table 1Table 1Overview of the Effect of Antibiotic-Loaded Liposomes Against Bacterial Infections.DrugMethodSizeTargetRemarksReferencesRifabutinDehydration–rehydration100–115 nmMRSA biofilmRifabutin-loaded liposomal formulations demonstrated superior efficacy compared to free vancomycin.[57]Tetracycline, Amoxicillin-270–340 nmMRSADrug-loaded liposomes enhanced the cellular uptake of antibiotics, thereby providing more effective treatment compared to their free forms.[58]ColistinThin layer hydration73–217 nm*Pseudomonas aeruginosa* infectionColistin-loaded cationic liposomes, with an encapsulation efficiency of 77%, had low MIC values against *Pseudomonas aeruginosa*.[59]AmoxicillinFilm hydration method210 nm*Staphylococcus aureus* infectionAmoxicillin-loaded PEG-
α-cyclodextrin-acrylamide-liposomes were incorporated into biocompatible hydrogels to prepare a wound dressing. The formulation demonstrated controlled drug release and antibacterial activity.[60]Piperacillin sodiumFilm hydration method95 nmAntibiotic resistance of clinical isolates of *Pseudomonas aeruginosa*Liposome-loaded Piperacillin exhibited superior antibacterial activity at a lower MIC value compared to its free form.[61]VancomycinFreeze–thaw method157 nmMethicillin-resistant *Staphylococcus aureus*At a 1:10^2^ dilution, free vancomycin failed to inhibit bacterial growth, whereas the liposome-loaded form achieved 100% inhibition.[62]AmpicillinSuperLip200 nm-Ampicillin-loaded liposomes were entrapped in alginate gels. This method resulted in enhanced encapsulation efficiency and improved polydispersity index values, indicating a more effective formulation.[63]AzithromycinProliposome164–187 nm*Chlamydia trachomatis*The formulations exhibited at least twofold higher activity compared to the free form against both clinical isolates and bacterial strains.[64]Ciprofloxacin and colistinThin film evaporation and sonication102.1–119.7 nmClinical isolates of *Pseudomonas aeruginosa* H131300444 and *P. aeruginosa* H133880624The formulation exhibited effective antibacterial properties against 
*P. aeruginosa*, a multidrug-resistant Gram-negative bacterium responsible for pulmonary infections, while showing no cytotoxic effects on A549 cells. However, the encapsulation efficiency for both drugs remained below 50%.[65]LevofloxacinFilm hydration method128 nm*Staphylococcus aureus*Levofloxacin-loaded liposome formulations were coated with chitosan (CS), which caused an increase in particle size, along with an enhancement in antibacterial activity.[66]


## 5. Therapy of *Mycobacterium tuberculosis* Infection

Airborne and primarily affecting the lungs, tuberculosis is a highly lethal disease caused by *Mycobacterium tuberculosis*. According to WHO data, tuberculosis affects approximately 10 million people annually and results in about 1.3 million deaths worldwide [67]. The treatment of this disease commonly involves rifampicin (RIF), isoniazid (INH), pyrazinamide (PZA), and ethambutol (EMB). For the treatment of multi-drug-resistant *M. tuberculosis*, the WHO has recommended the use of a mixture of moxifloxacin/levofloxacin, clofazimine (CLF), ethionamide/prothionamide, INH, PZA, and EMB for 4 months, and a combination of moxifloxacin/levofloxacin, clofazimine, PZA, and EMB for 5 months, with bedaquiline (BDQ) being used for the initial 6 months [61].

Above, we introduced (Section 2) some of the earliest studies on the use of liposome-encapsulated antibiotics in the therapy of *M. tuberculosis*. These studies have paved the way for more extensive studies with newer antibiotics and different liposome compositions. Because of the potential development of resistance to these drugs, novel, and more effective therapeutic strategies are needed [68].

For the treatment of tuberculosis, liposomes have been designed to encapsulate antibiotics such as RIF [69], INH [70], PZA [71], and EMB [72], either individually or in combination [73]. Additionally, potential compounds with anti- tuberculosis activity [19] have also been loaded into liposomes for therapeutic applications, including theranostic approaches that combine diagnosis of the pathologic lesions and the delivery of the therapeutic agent. Furthermore, increasing the particle size can influence the therapeutic effect. (Figure 2). In addition to their use in drug delivery for therapeutic purposes, liposomes can also be loaded with theranostic agents to enable early detection of disease and early identification of drug-related adverse effects, as well as to support the development of alternative treatment strategies. In this context, liposomes co-loaded with silver nanoparticles and rifampicin were synthesized, and their emission spectra were recorded using UV-spectrophotometry, revealing peaks at 565 and 590 nm [74]. In another study, PEGylated liposomes with an approximate size of 160 nm were formulated for theranostic applications in mycobacterial infections, encapsulating 66.9 ± 10.9% of rifampicin and 40.6 ± 8.7% of ofloxacin. These liposomes were radiolabeled with technetium-99m. In vivo biodistribution studies indicated that the radiolabeled liposomes predominantly accumulated in the spleen, liver, and kidneys. The formulated liposomal system demonstrated promising anti-tuberculosis efficacy [75].

Adams et al. [76] incorporated CLF in multilamellar DMPC:DMPG (7:3) liposomes using a lyophilization technique. As this antibiotic is highly lipid-soluble, an encapsulation efficiency of more than 95% was achieved. The free antibiotic at the maximum tolerated dose of 5 mg/kg of body weight was ineffective in reducing the CFU in a murine model of acute tuberculosis. A 10-fold-higher dose of liposomal CLF, given intravenously, was not toxic and reduced the CFU 2 to 3 log units in the liver spleen and lungs. Liposomal CLF resulted in no detectable CFU in the spleen or liver, and a 1- to 2-log reduction in the lungs in an established or chronic infection. When liposomal CLF was administered again, *M. tuberculosis* was cleared in all three tissues, indicating that this treatment was bactericidal.

Gangadharam et al. [77] reported that streptomycin encapsulated in sterically stabilized liposomes with prolonged circulation times have a therapeutic effect in beige mice infected with MAC. Using similar low clearance liposomal amikacin in dosages of 160, 80 and 40 mg/kg given iv three times a week to a murine model of *M. tuberculosis*, Dhillon et al. [78] showed that they were 2.4–5.0 times more active than free amikacin and about 6.7 times more active than streptomycin, with the non-liposomal drugs given intramuscularly five times a week. Deol and Khuller [79], Deol et al. [80], and Labana et al. [81] utilized O-stearoylaminopectin in the membrane of multilamellar liposomes composed of PC:chol:dicetylphosphate:PEG-DSPE, in an attempt to localize liposomes encapsulating INH and RIF in the lungs of *M. tuberculosis*-infected mice. These liposomes are indeed preferentially localized in the lungs rather than in the reticuloendothelial systems (RES) of normal and tuberculous mice. Although the liposomes were described as being “stealth,” the multilamellar nature of the liposomes, and hence their large size, probably precluded their avoidance of the RES. Nevertheless, the authors utilized a clever technique involving the pre administration of PC:chol (2:1.5) liposomes before the delivery of the lung-specific liposomes, which resulted in increased uptake in the lungs [80]. The liposomes mediated the sustained release of the drugs in plasma, lungs, liver and spleen over a 5–7 day period. The area under the curve values of the encapsulated drugs were greater than those for the free drugs. The liposomal drugs administered once a week for 6 weeks reduced significantly the bacterial CFU in the lungs, liver, and spleen of the mice compared to untreated controls.

Free CLF, at the maximum tolerated dose of 5 mg/kg was ineffective in reducing the CFU of *M. tuberculosis* in the spleen, liver and lungs of infected mice. A 10-fold-higher dose of liposome-encapsulated CLF given twice a week caused a significant reduction in live bacteria in all tissues, and without any toxic effects [76]. In established or chronic infection, there were no detectable CFU in the spleen or liver of infected mice, and 10–100-fold reduction in the lungs. In a highly significant observation, a second series of liposomal CLF administration cleared the bacteria in all three tissues.

RIF or INH incorporated in multilamellar liposomes composed of PC:chol:dicetyl phosphate:PEG-DSPE:O-stearoyl amylopectin were delivered intravenously to guinea pigs and shown to maintain drug levels in plasma for 4 to 7 days as well as in tissues, while the free drugs, given orally or intravenously were cleared within 10–12 h [82]. In animals infected with *M. tuberculosis* (H37Rv strain), liposomal drugs were administered once a week (7 doses), whereas the oral drugs were given daily for 46 days (10 mg/kg for RIF and 12 mg/kg for INH for both conditions). At the end of the experiment, there was a 1.7 log reduction in CFU in lung/spleen homogenates for both cases, even though there was a 6.6-fold difference in the total antibiotic delivered. This advantage may lead to a preference by patients for weekly liposomal delivery [82].

Multilamellar liposomes composed of DPPC:chol (7:2) incorporating pyrazinamide (PZA), and free PZA were administered subcutaneously to mice infected with *M. tuberculosis* at a dose of 25 mg/kg. Lung tissue was examined for bacterial CFU 30 days after the last treatment and showed a significant reduction from untreated controls both for twice weekly liposomal PZA and 6 days/week free PZA [83].

A palmitic acid derivative of isoniazid, 4-(5-pentadecyl-1,3,4-oxadiazol-2-yl)pyridine, was synthesized and incorporated into the membrane of liposomes composed of PC:chol [84], utilizing the reverse-phase evaporation method of Szoka and Papahadjopoulos [85], but without acknowledging the inventors. The molar lipid ratio was given as 0.3:0.7:1, without explicitly indicating the correspondence of the mol fractions. If the PC mol fraction was indeed 0.3 and the cholesterol mol fraction was 0.7, as the sequence suggests, it is difficult to imagine how such an excess of cholesterol can be incorporated in the membrane. The palmitic acid chain at a mol fraction of 1 most likely caused the micellization of the mixture. Nevertheless, these liposomes were tested in BALB/c mice infected with *M. tuberculosis* 2 months prior to the intratracheal treatment with 50 μg of the oxadiazole derivative for 1 month. Mice infected with the MDR strain were treated with 150 μg of the oxadiazole derivative for 2 months. In the former experiment, an 80% decrease in live bacilli were detected in the lungs. In the latter experiment, a decrease of 90% of live bacilli was observed [84]. This is clearly a promising result for the eventual treatment of patients infected with MDR strains, if these liposomes can be scaled up and are properly characterized.

Because of the potential development of resistance to the drugs discussed above, novel and more effective therapeutic strategies are needed urgently [68].

Formulations of various drugs prepared using liposomal delivery systems not only enhance therapeutic efficacy but also serve as promising platforms for the development of vaccines against tuberculosis infections [86]. Tuberculosis (TB) vaccines can be categorized into three main groups. The first group comprises live or attenuated recombinant Bacille Calmette–Guérin (BCG) vaccines, which are known for their high immunogenicity. The second group includes viral vector-based vaccines and adjuvanted subunit vaccines. The third group consists of vaccines derived from whole-cell or fragmented components of *Mycobacterium* species [87].

One of the most critical steps in the development of a liposomal TB vaccine is the selection of appropriate lipids. The choice of cationic lipids, zwitterionic lipids, and cholesterol plays a key role in the formulation. During the optimization phase, liposome preparation is carried out, and efficient incorporation of the antigen into the liposomal structure is essential. The synthesized liposomes must be characterized in terms of size, zeta potential, polydispersity index, and stability.

In vitro evaluations should include cellular uptake, cytotoxicity, dendritic cell (DC) activation, cytokine production, and T-cell activation. Upon completion of these stages, the study proceeds to in vivo experiments.

A novel vaccine strategy has been explored by combining Poly:IC adjuvant with liposomes containing the Ag85B and ESAT-6 antigens [88].

Cationic liposomes can trigger antigen-specific immune responses and, due to their intrinsic immune-stimulatory properties, promote the maturation of DCs while inducing both CD4^+^ Th1 and CD8^+^ T-cell responses. Cationic liposomal formulations were developed using the Ag85B-ESAT6-Rv2034 (AER) fusion protein antigens for potential use as a tuberculosis vaccine [89]. Formulations prepared with different cationic lipids and ratios exhibited particle sizes ranging from 86 to 230 nm, with polydispersity indices between 0.22 and 0.32. All formulations demonstrated zeta potentials greater than +16 mV. Evaluation of cellular uptake, viability, and cytokine production by human monocyte-derived dendritic cells revealed that the formulations were biocompatible. Among them, EPC:cholesterol:DOPC and DOTAP:cholesterol:DOPC formulations demonstrated superior cellular uptake compared to the others.

The formulation AER/EPC:cholesterol:DOPC at a ratio of 2:1:2, which was identified as the most effective in upregulating surface activation markers, was further tested for T-cell recognition. Notably, this formulation led to a significant increase in the proportion of IFNγ^+^ CD154^+^ double-positive T cells compared to empty liposomes. Furthermore, analysis of cytokine induction by AER/EPC:cholesterol:DOPC formulations prepared at varying ratios revealed enhanced levels of CCL3, CCL4, CXCL10, and CCL11 [89].

To induce a stronger immune response, fusion proteins such as Hspx, PPE44, and EsxV have been loaded into liposomes for the development of a tubeculosis vaccine [90].

In vivo studies of the developed liposomal vaccine demonstrated that the induction of CD4^+^ and CD8^+^ T-cell responses in immunized mice plays a critical role in the control of *M. tuberculosis* infection. pH-sensitive liposomes composed of DOPC:DOPE:DOBAQ:EPC at a molar ratio of 3:5:2:4 were formulated as a delivery system for the AER fusion protein (Ag85B-ESAT6-Rv2034), along with the adjuvants CpG and MPLA. This formulation significantly enhanced both CD4^+^ and CD8^+^ T-cell responses. Moreover, the vaccine elicited robust antigen-specific antibody titers, further supporting its potential as a promising protective candidate against tuberculosis [91]. One of the major limitations of liposome-based vaccines is their limited stability. In addition to the high cost of cationic lipids and challenges associated with large-scale manufacturing, these systems often exhibit restricted loading capacity for both antigens and adjuvants, which may compromise vaccine efficacy and formulation efficiency [92].

The FDA approval of liposome-based drug formulations and the widespread utilization of this nanocarrier system highlight its significance in advancing next-generation anti-tuberculosis treatment approaches [92]. Thanks to remarkable advancements in liposome technology, the development of inhalable antibiotic-loaded liposomes, in addition to those administered via the intravenous [93] or oral route [40], has been made possible. Additionally, targeted drug delivery systems help reduce the degradation of drugs in the body compared to their free forms, allowing for the administration of lower doses while maintaining therapeutic efficacy. As a result, the toxic effects of the drugs are minimized, leading to a more effective treatment.

These formulations hold potential for clinical applications and patent acquisition, ultimately making them available for human use [94]. This characteristic provides a significant advantage in the treatment of tuberculosis, which is primarily localized in the lungs [71,95]. Although oral administration of rutin-based drugs is more frequent and cost-effective, their low gastrointestinal absorption and rapid hepatic first-pass metabolism necessitate high-dose administration. Therefore, parenteral and pulmonary delivery of rutin-based drugs offers a higher bioavailability, as they bypass the first-pass metabolism. *M. tuberculosis* evades macrophage-mediated bactericidal mechanisms by inhibiting the formation of phagolysosomes [96]. Liposomes can be administered via inhalation depending on their size. Liposomes with a particle size ranging from 0.1 to 2 µm can reach the alveoli, whereas those larger than 15 µm cannot penetrate the respiratory barrier [97].

For targeted applications, if the aim is to reduce the immunogenicity of liposomes and decrease their uptake by the mononuclear phagocyte system, their surface can be modified with PEG [98]. PEGylated liposomes have been developed for the co-delivery of antitubercular drugs and TGF-β1 siRNA for tuberculosis treatment. The formulated liposomes exhibited minimal toxicity to human macrophages while demonstrating good selectivity [99].

The conjugation of various polysaccharides, including chitosan, dextran, and fucoidan, to the surface of liposomes can enhance their stability while also imparting mucoadhesive properties. Additionally, these formulations not only improve cellular uptake but also interact with carbohydrate receptors expressed on the surface of macrophages [100]. Table 2 summarizes liposomal formulations used to treat tuberculosis, some of which are described only in the table, and are not discussed in the text. Fucoidan-based surface modifications have been applied to enhance the activity of usnic acid-loaded liposomes in tuberculosis treatment [101]. The use of cationic pH-sensitive liposome-based delivery systems facilitates the passage through the endosomal membrane under low pH conditions, enabling the release of the encapsulated content into the cytosol. This prevents the degradation of delivered antigens and, consequently, enhances cytotoxic CD8^+^ T-cell responses [91]. In tuberculosis treatment, to overcome local and systemic toxicity, liposomes encapsulating anti-tuberculosis drugs can be surface-modified with macrophage-specific ligands. To facilitate the delivery of RIF-loaded liposomes to alveolar macrophages, liposomal formulations coated with maleylated bovine serum albumin and O-stearoyl amylopectin were developed. Based on drug localization index data, ligand-functionalized drug-loaded liposomes exhibited a 1.4- to 3.5-fold higher localization compared to non-ligand-modified counterparts. These findings indicate that the ligand-modified liposomes enable rapid drug delivery to the lungs and achieve high drug concentrations at the target site [102]. 

Fucosylated liposomes with an average diameter of approximately 70 nm have been developed to improve the pulmonary bioavailability of antituberculosis agents such as bedaquiline. These liposomes specifically target the macrophage mannose receptor (CD206), enabling selective intracellular delivery. Compared to the free drug, the targeted liposomal formulation demonstrated superior antitubercular activity and a reduction in systemic side effects [103].

The physicochemical properties of dry powder inhalers used during treatment are of critical importance. In particular, for potent anti-*M. tuberculosis* agents such as pretomanid, trehalose is employed as an excipient to enhance drug solubility. To protect the hygroscopic trehalose, spray drying was performed in combination with L-leucine. The use of L-leucine at a nearly 1:1 mass ratio resulted in improved entrapment efficiency. The particle sizes of the resulting liposomal formulations ranged from 130 to 300 nm and were found to be safe for both broncho-epithelial cell lines and alveolar macrophages. Furthermore, the formulated liposomes exhibited superior antimycobacterial activity compared to the free drug [104].

Glutathione deficiency increases susceptibility to *M. tuberculosis* infection. In in vivo models infected with *M. tuberculosis*, glutathione—known for its antioxidant and immunomodulatory properties—was shown to reduce *M. tuberculosis* survival in the liver and spleen. Additionally, glutathione treatment decreased oxidative stress in these tissues and led to a reduction in IL-6 levels, while increasing the levels of IFN-γ and TNF-α, indicating a shift toward a more effective immune response against the pathogen [105].

A new therapeutic strategy has emerged in which mycobacteriophages were loaded into liposomes for the treatment of tuberculosis, presenting an innovative approach to combat the disease [106]. We have previously reviewed the therapeutic potential of bacteriophages [107]. Table 2 summarizes liposomal formulations used to treat tuberculosis (some of these are described only in the table and are not discussed in the text).
antibiotics-14-00728-t002_Table 2Table 2New Liposomal Strategies Against Tuberculosis.DrugMethodSizeRemarksReferencesCationic pH-sensitive liposomeThin-film hydration165 nmpH-sensitive cationic liposomes formulated with the Ag85B-ESAT6-Rv2034 fusion antigen and CpG and MPLA adjuvants have been shown to induce potent polyfunctional CD4^+^ and CD8^+^ T-cell responses. Additionally, an increase in CD69^+^ B-cell sub-populations was observed.[91]BedaquilineThin-film hydration70 nmFucosylated liposomes have been developed to improve the pulmonary bioavailability of antituberculosis agents such as bedaquiline. These liposomes specifically target the macrophage mannose receptor (CD206), enabling selective intracellular delivery. Compared to the free drug, the targeted liposomal formulation demonstrated superior antitubercular 
activity and a reduction in systemic side effects.[103]PretomanidSpray drying130–300 nmThe physicochemical properties of dry powder inhalers used during treatment are of critical importance. In particular, for potent anti-*M. tuberculosis* agents such as pretomanid, trehalose is employed as an excipient to enhance drug solubility. To protect the hygroscopic trehalose, spray drying was performed in combination with L-leucine. The use of L-leucine at a nearly 1:1 mass ratio resulted in improved entrapment efficiency. The particle sizes of the resulting liposomal were found to be safe for both broncho-epithelial cell lines and alveolar macrophages. Furthermore, the formulated liposomes exhibited superior antimycobacterial activity compared to the free drug.[104]Glutathione--Glutathione deficiency increases susceptibility to *M. tuberculosis* infection. In in vivo models infected with *M. tuberculosis*, glutathione—known for its antioxidant and immunomodulatory properties—was shown to reduce *M. tuberculosis* survival in the liver and spleen. Additionally, glutathione treatment decreased oxidative stress in these tissues and led to a reduction in IL-6 levels, while increasing the levels of IFN-γ and TNF-α, indicating a shift toward a more effective immune 
response against the pathogen.[105]Lytic Mycobacteriophage D29Thin-film hydration800 nmThe lytic mycobacteriophage D29 was successfully formulated into a liposomal preparation and based on the results of both in vitro and in vivo studies, the formulated liposomal form of mycobacteriophage D29 demonstrated a pronounced lytic effect in both the in vitro granuloma model and the tuberculosis infection model established in C57BL/6 mice.[106]Anionic and neutral
liposomes
-For improved pulmonary TB treatment, ID93 plus GLA-containing liposomal adjuvant formulations were developed. However, the anionic or neutral liposome + QS-21 liposomal formulations did not result in a significant reduction in *M. tuberculosis* bacterial load. Nevertheless, these formulations were observed to induce distinct immune responses.[108]Moxifloxacin loaded liposome-siderophore
conjugatesFilm hydration
technique200 nmLiposome formulations with a spherical shape had an encapsulation efficiency of 46% and demonstrated anti-TB activity with a MIC value of 0.32 µg/mL.[109]Saquinavir Thin-film hydration116 nmIn the treatment of multidrug- and extensively drug-resistant *M. tuberculosis* strains, negatively charged Saquinavir-loaded liposomes were shown to enhance intracellular killing activity by human macrophages.[110]RifampicinThin-Layer Evaporation117 nmThe prepared formulation demonstrated a greater reduction in intracellular *M. abscessus* viability compared to the free form of the drug.[111]Oral liposomal glutathione supplementation--Commercially available liposomal glutathione supplementation (L-GSH) has been shown to reduce oxidative stress in patients with type 2 diabetes mellitus (T2DM). In vitro models have demonstrated their ability to decrease intracellular mycobacteria.[112]Rifampicin and
isoniazidLipid film hydration, sonication and extrusion-Antibiotic loaded, polyorganophosphazene-arginine-grafted liposomes exhibited a 73% RIF and 80% IZN release at endosomal pH. The liposomes demonstrated a dose-dependent inhibition of *M. tuberculosis* growth in culture medium.[113]N′-Dodecanoylisonicotinohydrazide
∼130 nmFor use in localized tuberculosis treatment, the Isoniazid derivative *N′dodecanoylisonicotinohydrazide*, a commonly used agent in tuberculosis therapy, was loaded into liposomes. PLGA-PEG-PLGA systems were incorporated to develop thermosensitive and self-healing hydrogel systems. Data obtained from in vivo microdialysis studies demonstrated the rapid release of the drug into the synovial fluid.[114]Coumaran (2,3-dihydrobenzofuran) derivatives—TB501 and TB515—Thin film hydration∼60 nmThe liposome formulation prepared with TB515 exhibited high encapsulation efficiency. Multicomponent pH-sensitive stealth liposomes encapsulating TB501 were highly effective against *M. tuberculosis* in macrophage cell lines.[98]Zn-phthalocyanineEthanol injection134 nmZnPC-loaded liposomes, prepared for the treatment of Rifampicin-Isoniazid-resistant *M. tuberculosis* strains, achieved a 99.9% cell death rate in vitro through photodynamic therapy (PDT).[115]IsoniazidThin-film hydration37–45 nmBiocompatible hydrogenated soy phosphatidylcholine-phosphatidylglycerol liposomes were developed as isoniazid carriers. The encapsulation efficiency was determined using UV and Laser Transmission Spectroscopy.[116]Glucopyranosyl lipid adjuvant (GLA) and the
experimental tuberculosis vaccine, ID93, composed of four *M. tuberculosis* antigensThin-film hydration, sonication,
homogenization50–87 nmFor use as a vaccine in the treatment of *Mycobacterium tuberculosis*, formulations containing a TLR4 agonist (GLA) and QS21, in combination with ID93, were developed. In an in vivo model, these formulations demonstrated a reduction in bacterial load in the lungs of mice infected with *M. tuberculosis*. Clinical studies involving human participants are ongoing to evaluate the safety, tolerability, and immunogenicity of the developed formulations.[117]Artemisone, 
Clofazimine and 
DecoquinateThin-film hydration147, 482,
253 nmDrug-loaded liposomes, synthesized in various sizes, exhibited 32–42% inhibition of *M. tuberculosis* growth in culture medium. By contrast, drug-free liposomes induced only 12% inhibition.[118]Isoniazid-conjugated Phthalocyanine“Heating Method”150–650 nmA complex of γ-Cyclodextrin with Isoniazid-conjugated Phthalocyanine, was incorporated into crude soybean lecithin liposomes using a simple and measurable heating method. This pH-sensitive formulation exhibited 100% drug release at pH 4.4, while releasing only 40% at pH 7.4, demonstrating its potential 
applicability in targeted therapies.[119]Isoniazid and RifampicinReverse Phase
Evaporation332–361 nmLiposomal formulations loaded with anti-TB drugs Isoniazid, Rifampicin, and their combination were developed for inhaled therapy. Isoniazid formulations exhibited a faster release compared to Rifampicin formulations, while their encapsulation efficiencies were found to be similar.[120]


## 6. Therapy of Non-Tuberculous Mycobacterial Infections

The most common appearance of non-tuberculous mycobacteria (NTM) infections is lung disease [121]. These bacteria, which include *M. avium-M. intracellulare* Complex (MAC)*, M. abscessus* and *M. chimaera*, are present in the environment and can cause pulmonary infections in immunocompromised individuals or persons who have lung damage. NTM may present as biofilms attached to the alveolar wall or intracellularly in monocytes and macrophages.

Cynamon et al. [122] reported that liposome-encapsulated gentamicin in the beige mouse model of disseminated MAC infection was more effective than amikacin alone, and that the combination of this preparation with rifapentine was more active than either treatment by itself, which would be expected. By contrast, the addition of clarithromycin to the liposomal gentamicin produced results similar to those of either agent alone. Clofazimine, however, improved the activity of the liposomal gentamicin in the spleen, and ethambutol enhanced the antimycobacterial activity in the liver. These free antibiotics added to the effect of the liposomal gentamicin in the lungs.

Gangadharam et al. [77] utilized two liposome types with prolonged circulation: PEG:distearoylphosphatidylethanolamine (DSPE): distearoylphosphatidylcholine (DSPC):cholesterol (chol), or phosphatidylinositol (PI):DSPC:chol liposomes, encapsulating streptomycin and administered twice weekly were bactericidal to MAC in the spleen. PI:DSPC:chol:streptomycin liposomes were bactericidal in the liver. PEG-DSPE:DSPE:chol: streptomycin liposomes were not bactericidal in the lungs, but they reduced the level of MAC infection by more than three orders of magnitude compared with the CFU in untreated controls.

We investigated the therapeutic efficacies of liposome-encapsulated streptomycin and ciprofloxacin against MAC in human peripheral blood monocyte/macrophages. Liposomal streptomycin was 3-fold more effective than the free antibiotic, in the concentration range 10–50 µg/mL [123]. Liposome-encapsulated ciprofloxacin, even at 0.1 µg/mL was at least 50 times more effective than the free drug in killing intracellular bacteria. At 5 µg/mL, liposomal ciprofloxacin reduced the colony forming units (CFU) by more than 1000-fold at the end of the 7-day incubation period, suggesting that liposome-encapsulated fluoroquinolones may be effective against MAC infections in vivo. It is surprising that this highly effective formulation has not been investigated further in animal models of MAC infection. Showing the feasibility of utilizing this liposomal antibiotic, Chono et al. [124] have found that pulmonary administration of ciprofloxacin in multilamellar liposomes sized to 1 µm diameter resulted in the localization of the antibiotic in alveolar macrophages without causing cellular toxicity.

In the first exploratory human trial of a liposomal antibiotic, gentamicin encapsulated in egg PC multilamellar liposomes in the size range 1.2–10 µm (designated as TLC-G65, developed by The Liposome Company; now Elon Pharmaceutical Technologies, Dublin, Ireland), was administered intravenously twice a week for 4 weeks to AIDS patients with MAC bacteremia, at three different doses [125]. The bacterial colony counts in blood was reduced by ≥75% in all the groups, and no drug resistance was developed during the 39-day study period.

The uptake of ofloxacin and clarithromycin encapsulated in multilamellar egg PC:chol:dicetylphosphate (7:2:1) liposomes by human peripheral blood macrophages, administered at clinically achievable concentrations, was enhanced compared to the free drugs [126]. The liposomal antibiotics were significantly more effective in killing intracellular MAC than the unencapsulated drugs, clarithromycin being more effective than ofloxacin. efficacy of clarithromycin, either in the free or liposome-entrapped form, was markedly higher than that of ofloxacin. When ethambutol was added to either liposomal drug, the efficacy was greater than that of either treatment alone. The authors suggested that combination therapy with ethambutol could further enhance the efficacy of liposomal clarithromycin [126]. This study was extended by the same laboratory by utilizing both antibiotics and granulocyte-macrophage colony-stimulating factor (GM-CSF) in liposome-encapsulated form [127]. The anti-mycobacterial effect of the liposomal cytokine was higher by two orders of magnitude compared to the free form. When GM-CSF was added to azithromycin and low concentrations of ofloxacin and clarithromycin that are achieved therapeutically in vivo, the antimycobacterial activity was enhanced compared to the effects of the agents alone. Thus, the activation of macrophages by cytokines may be a useful adjunct in the antibiotic treatment of intracellular MAC infection.

Straubinger and co-authors encapsulated ciprofloxacin with >90% efficiency by utilizing both a pH- and potential-gradient across the membrane of liposomes composed of DSPC:DSPG:chol (1:1:1) and containing ammonium sulfate at pH 2.5, that facilitated the passive transport of ciprofloxacin into the interior aqueous space and its entrapment [128]. This technique is usually called “remote-loading” and was developed by the Cullis and Barenholz laboratories to entrap doxorubicin [56,129]. Ciprofloxacin encapsulated in highly negatively charged DSPG:chol (1:1) liposomes had the highest accumulation in MAC-infected J774 murine macrophages and the maximal antimicrobial effect (43-fold higher than the free drug). Azithromycin incorporated in the membrane of DSPG:chol (1:1) liposomes prepared by a lyophilization, rehydration and sonication method inhibited the intracellular growth of MAC 41-fold more effectively than free azithromycin at the same concentration.

Small (54–65 nm diameter) unilamellar liposomes encapsulating amikacin, designated “VS107,” were administered intravenously to MAC-infected mice three times a week, and after 51 days reduced the CFU by 100-fold in the liver and spleen compared to free amikacin, and by 6 log units compared to untreated controls [130].

Twice weekly intravenous injection of amikacin liposomes over a 3-week period in MAC-infected immunodeficient mice reduced the mycobacterial CFUs by 3 to 4 logs in the spleens and livers and continued to stay low in the liver [131]. When mice were treated in the chronic stage of infection, granulomatous inflammation in the liver was reduced, and even very sick mice gained weight and lived 4 months longer than untreated animals. Pulmonary MAC infection, however, did not respond as well to intravenous liposomal amikacin treatment. This observation most likely directed research efforts to pulmonary delivery of liposomal antibiotics.

As an approach to this problem, Zhang et al. [132] aerosolized amikacin-liposomes (DPPC:chol (2:1)) into rats infected with non-tuberculous mycobacteria (NTM; in this case, *M. avium* subsp. *hominissuis*), and observed a large increase in the mean area under the concentration-time curve in the lungs (42-fold) and lung macrophages (274-fold) compared to intravenous free amikacin. These early studies have been translated into clinical trials of amikacin liposome inhalation suspension (“ALIS”), which we describe in the next section (Section 7).

In the treatment of MAC infections, in addition to conventional liposomal formulations of antibiotics, combinations with antimicrobial peptides have been explored to harness potential synergistic effects. One such peptide, the cyclic peptide [R4W4], composed of four arginine and four tryptophan residues, exhibits broad-spectrum antimicrobial activity [133]. The liposomal formulation of [R4W4] alone had an average particle size of 156 nm, while the incorporation of rifamycin or azithromycin into the liposomes resulted in an increase in particle size. These combination formulations significantly reduced the intracellular viability of MAC, indicating enhanced antibacterial efficacy [133].

Fucosylated liposomes encapsulating bedaquiline were processed into dry powder inhalation formulations using a spray-drying technique. The resulting powders exhibited a high fine particle fraction, making them suitable for pulmonary administration. In vitro evaluation using *M. abscessus*-infected THP-1 macrophages revealed that the fucosylated liposomal formulation achieved more efficient bacterial clearance than the free drug, underscoring its potential as an advanced pulmonary delivery system [134].

In a study to establish whether prolonging the intervals between drug administration in the therapy of chronic, lethal MAC infection in a murine model, Leitzke et al. [135] found that liposome-encapsulated amikacin produced high and sustained drug levels in infected tissues with once-weekly and once-monthly treatments.

Rifabutin incorporated into liposomes composed of PC:phosphatidylserine (7:3) and administered intravenously to mice infected with the virulent *M. avium* strain P1581 resulted in a significant enhancement of antimicrobial activity compared to that of free rifabutin [136].

The antibiotic clofazimine was shown to be much less toxic to host cells when encapsulated in liposomes, enabling the administration of higher doses [137,138]. Administration of liposomal clofazimine resulted in a significant reduction in the CFU of MAC in the liver, spleen, and kidneys.

Resorcinomycin A encapsulated in DMPC:phosphatidylinositol (9:1) liposomes reduced the CFU of MAC in murine peritoneal macrophages by 50–93%, while the free antibiotic caused a 33–62% reduction, when applied in the concentration range 6–50 µg/mL [139]. This antimycobacterial activity was maintained for 7 days after treatment, whereas free resorcinomycin A was inactive 3 days after treatment.

The Bakker-Woudenberg laboratory examined the efficacy of combination therapy involving the recently adopted antibiotics against MAC infection with amikacin delivered in sterically stabilized liposomes with prolonged circulation [140]. Although daily clarithromycin killed most of the microorganisms in the lung, liver, spleen, inguinal and mesenterial lymph nodes, after 24 weeks of treatment, a substantial amount of bacteria persisted in the infected organs. The addition of ETH or RIF to clarithromycin did not enhance significantly the outcome of the treatment. The inclusion of liposomal amikacin in the initial phase of therapy, however, resulted in the complete elimination of the bacteria in all the infected organs within 12 weeks of treatment. Thus, it was possible to reduce the entire treatment duration to 12 weeks.

In a novel approach, liposomal (DPPC:chol (2:1)) amikacin was aerosolized to treat established pulmonary *M. avium* subsp. *hominissuis* infection in mice. The animals that received 1 h daily amikacin via inhalation and 2 h every other day cleared more *M. avium* than those receiving intraperitoneal free amikacin, even though they received 32% less total amikacin [141]. At 10 µg/mL, liposomal amikacin was also significantly more effective in killing *M. avium* inside differentiated THP-1 macrophages compared to the same dose of free amikacin. Similar results were obtained for intracellular *M. abscessus* subsp. *bolletii* infection.

*Mycobacterium abscessus* pulmonary infections present a challenging disease, because the microorganism is resistant to many antibiotics. Rinaldi et al. used liposomes composed of DPPG:hydrogenated soy PC (1:1), with RIF in the membrane phase, and treated differentiated THP-1 macrophages infected with *M. abscessus* [111]. RIF in liposomes was more effective in reducing the mycobacterial load than the free drug, particularly at the higher concentrations (96 µM).

Several lipid-based nanocarrier systems have been explored for the treatment of *M. abscessus* infections, with liposomes being the most extensively studied. For example, liposomes encapsulating methyl tris diazeniumdiolate, a nitric oxide (NO)-releasing prodrug, exhibited significantly enhanced antimicrobial and antibiofilm activities compared to the free drug. This improvement was attributed to increased intracellular uptake facilitated by the liposomal carrier, resulting in elevated intracellular NO concentrations responsible for bacterial killing [142]. In another approach, phosphatidylserine-based liposomes were developed to modulate host immune responses. These liposomes promoted phagosome acidification and enhanced reactive oxygen species (ROS) production. Furthermore, they suppressed NF-κB activation and reduced TNF-α levels in monocyte-derived macrophages infected with *M. abscessus*, suggesting a dual antimicrobial and anti-inflammatory effect [143].

Mucoadhesive liposomal formulations have also been investigated for improved drug delivery. RIF-loaded liposomes were coated with chitosan or ε-poly-L-lysine—two polymers with known antimicrobial properties. Coating with chitosan increased liposome size significantly, whereas ε-poly-L-lysine-coated liposomes demonstrated superior cellular uptake and more potent intracellular antimicrobial activity in infected macrophages [144].

## 7. Clinical Studies on Liposomal Amikacin

Numerous preclinical studies have investigated the liposomal formulations of various antibiotics, including amikacin. However, clinical studies in this area remain limited. One of the main reasons for this limitation is the inherent challenges associated with liposomal drug delivery systems, such as stability issues, cold-chain storage requirements, high production costs, and manufacturing complexity. In particular, large-scale production demands strict sterility controls and advanced manufacturing processes, all of which significantly increase the overall cost.

Moreover, tuberculosis-specific indications necessitate extensive safety and efficacy data. Considering the complexity and duration of tuberculosis treatment regimens, these factors further complicate the clinical translation of liposomal antibiotic formulations.

Despite these challenges, a major milestone was achieved in 2018 when the FDA approved amikacin liposome inhalation suspension for the treatment of MAC lung disease in patients with limited therapeutic options. This product, marketed as Arikayce^®^, represents a significant advancement in the use of liposomal antibiotics for the treatment of pulmonary infections. Since its approval, numerous clinical studies have been conducted on this specific formulation, highlighting its therapeutic potential and paving the way for further development in this field.

A retrospective analysis of 17 patients undergoing treatment with amikacin-liposomes (“Amikacin Liposomal Inhalation Suspension,” ALIS) for NTM lung infection showed that at 6 months, 86% of the patients had clinical, microbiological, and radiological improvement. Twenty-five percent of the treated patients, some of whom were coinfected with *M. abscessus*, relapsed after the therapy was completed [145].

Clinical studies evaluating liposomal formulations of amikacin and gentamicin for the treatment of *M. abscessus* and MAC, although limited in number, have nevertheless been conducted. In cystic fibrosis (CF) and non-CF patients, amikacin liposome inhalation suspension (ALIS) was administered to 26 individuals alongside standard antibiotic therapy, and its safety and efficacy were evaluated. ALIS proved beneficial in both CF and non-CF populations with *M. abscessus* lung disease. Importantly, beyond sputum culture conversion, radiological stabilization or improvement was achieved in 19 of 26 (73%) patients, and pulmonary function stabilization or improvement was observed in 10 of 20 (50%) patients [146].

Patients with refractory MAC lung disease received guideline-based therapy (GBT) in combination with 590 mg of once-daily ALIS for 12 months. No new safety signals were observed with up to 20 months of cumulative ALIS exposure [147].

For pharmacokinetic evaluation of ALIS, fifty-three patients with sputum cultures positive for MAC who were receiving guideline-based therapy were administered once-daily ALIS at a dose of 590 mg. Analysis of serum and urine samples from these patients revealed amikacin concentrations that were lower than those previously reported following parenteral administration [148]. To assess the safety of ALIS, 336 patients with refractory MAC lung disease were enrolled: 224 received once-daily ALIS (590 mg) in combination with GBT, while 112 received GBT alone. The addition of ALIS led to a significantly higher rate of culture conversion at month 6. Furthermore, ALIS was generally associated with mild to moderate adverse events [149].

Among 59 Japanese patients with refractory MAC pulmonary disease—over 85% of whom were women—48 were randomized, 34 received ALIS + GBT and 14 received GBT alone. No deaths were observed. In the ALIS + GBT cohort, culture conversion occurred in over 25% of patients, whereas no culture conversion was detected in the GBT-only group. Treatment-emergent adverse events were similar across both treatment arms [150].

In a phase II study of ALIS in patients with treatment-refractory pulmonary nontuberculous mycobacterial disease (MAC or *M. abscessus*), 89 patients were enrolled, 44 of whom received ALIS. The ALIS-treated cohort exhibited a higher rate of sputum culture conversion. Specifically, a greater proportion of patients in the ALIS arm achieved negative sputum cultures by Day 84 (14 of 44 [32%]) compared to placebo (4 of 45 [9%]). Some patients discontinued therapy due to respiratory adverse events [151].

The Cipolla laboratory at Insmed Inc. studied the batch-to-batch variability of the ALIS formulation and the Lamira^®^ nebulizer system (PARI Pharma GmbH, Starnberg, Germany) that delivers the liposomes [152]. Three batches of ALIS at different lipid concentrations were tested with nine inhalation devices with different hole geometries, nebulization rates, and aerosol droplet size distribution. The mean liposome size after nebulization was between 269 and 296 nm and was similar to the samples that did not go through this process (276–292 nm). The nebulization process caused the leakage of about 35% of the amikacin from the liposomes. (range: 33.8% to 37.6%). There was no statistically significant difference in the aerosol particle size distributions generated. The emitted dose of amikacin was between 80.2% and 89.3% of the loaded dose. Thus, the aerosolization procedure generated relatively homogeneous liposomes [152].

Identifying potential complications following ALIS therapy is essential to assess the safety of the drug in patients. In individuals with refractory MAC pulmonary disease, ALIS treatment for six months facilitated sputum culture conversion in more than 50% of cases. However, chest CT findings did not always correlate with sputum culture conversion, highlighting the complexity of evaluating treatment response solely based on radiological imaging [153]. A patient with MAC pulmonary disease who had undergone single-lung transplantation was treated with ALIS therapy, and no recurrence of MAC disease was observed for 15 months following treatment [154].

ALIS has shown therapeutic potential, particularly in non-tuberculous mycobacterial pulmonary disease (NTM-PD). However, despite its targeted delivery advantages, emerging data indicate that ALIS therapy may still lead to the development of amikacin resistance and is associated with a relatively high frequency of drug-induced lung injuries. Clinical evidence has highlighted specific resistance mechanisms associated with ALIS use. In two patients with MAC pulmonary disease (MAC-PD) who developed resistance during therapy mutations in the 16S ribosomal RNA (rRNA) gene —known to confer high-level aminoglycoside resistance—was detected [155]. In one case, a sputum sample collected after 32 weeks of ALIS therapy exhibited a minimum inhibitory concentration (MIC) of 256 μg/mL, suggesting that prolonged exposure to inhaled amikacin may contribute to resistance development [155].

In addition to resistance, ALIS-related pulmonary toxicity has also been reported. For instance, a 74-year-old female patient with NTM-PD developed a persistent cough 59 days after initiating ALIS. Bronchoscopic evaluation confirmed findings consistent with organizing pneumonia. Following discontinuation of ALIS and transition to intravenous amikacin therapy, clinical improvement was observed [156].

Clinically, ALIS has shown promise for *M. abscessus* pulmonary disease. In a 12-month clinical study, patients received 590 mg ALIS as an adjunct to multidrug therapy. The treatment resulted in few serious adverse events, and nearly half of the patients achieved sputum culture conversion, demonstrating its clinical potential [157].

The reports we have summarized in this section indicate that inhalational delivery of liposomal antibiotics can be highly efficacious in the treatment of mycobacterial lung disease, as long as dosing, lipid composition and liposome size distribution are controlled.

## 8. Lipidic Nanoparticles as Alternatives to Liposomes

As alternatives to liposomes, solid lipid nanoparticles (SLNs) and niosomes have also been employed in the treatment of mycobacterial infections. Solid lipid nanoparticles are particularly notable for their biocompatibility, biodegradable lipid core, ease of surface modification, large surface area, and particle size typically ranging between 100 and 400 nm. Due to the modifiable nature of their surface, these drug delivery systems offer broad applicability in therapeutic contexts. Chae et al. demonstrated that curcumin-loaded mannosylated solid lipid nanoparticles (Man-CUR SLNs) exhibited significantly stronger antibacterial activity against *M. intracellulare*-infected macrophages compared to the free form of curcumin [158].

Inhalable solid lipid nanoparticles loaded with levofloxacin were developed, featuring enhanced aerosol properties, safety, and efficient mucus penetration, with an encapsulation efficiency ranging between 40 and 50% [159]. These nanoparticles exhibited superior antibacterial activity compared to free levofloxacin. Moreover, the PEG-modified version of the formulation provided improved mucus penetration [159].

Another lipid-based nanoparticle system, niosomes, which are suitable for delivering both hydrophilic and hydrophobic drugs, are also under investigation. Fucoidan-functionalized niosomes loaded with silver nanoparticles (AgNPs) effectively reduced the intracellular survival of *M. abscessus* in macrophages, indicating that such nanoformulations may serve as viable alternatives to conventional therapies [160]. A microfluidic-based pH-sensitive niosome formulation loaded with lactoferricin, approximately 172 nm in size and possessing a zeta potential of −70 mV, had an encapsulation efficiency of 75.6%. This formulation exhibited low toxicity and pH-dependent drug release behavior, with 80% of the drug released at acidic pH and approximately 50% at physiological pH (7.4). The lactoferricin-niosome formulation showed enhanced anti-mycobacterial activity against both extracellular and intracellular *Mycobacterium tuberculosis* [161].

## 9. Future Directions, Challenges, and Limitations

The pharmaceutical industry is urged to develop additional liposomal antibiotics for the treatment of refractory lung infections, as well as disseminated mycobacterial infections. The experience of amikacin liposomes should have paved the way for the expeditious approval of alternative and novel antimycobacterial agents.

Liposomes, in addition to being biocompatible and undergoing natural degradation, offer the advantages of easy synthesis and modification with various agents. Their ability to encapsulate both hydrophilic and hydrophobic first-line anti-tuberculosis drugs, as well as other potential therapeutic compounds, further enhances their versatility. Additionally, liposomes can be utilized as vaccine carriers against tuberculosis. In the near future, formulations developed with liposomal technology hold promise for patentability, while providing effective solutions in tuberculosis treatment, ultimately improving patients’ quality of life and minimizing the adverse impacts of the disease on public health. One of the major challenges associated with liposomes is drug leakage, which can significantly impact their efficacy. Nevertheless, this challenge can be met by creative design of liposomes with greater stability, possibly by the employment of lipids from marine invertebrates that have longer acyl-chains, but with increased double bonds, conferring favorable thermotropic properties [162,163]. The high production cost of this carrier system represents a significant limitation. One potential strategy to reduce manufacturing expenses is the use of naturally sourced phospholipids (e.g., soybean lecithin or egg-derived phosphatidylcholine) instead of high-cost synthetic phospholipids. In addition, microfluidic systems can be utilized to enhance production efficiency by minimizing material waste, reducing solvent and energy consumption, and enabling precise control over formulation parameters. This approach also reduces batch-to-batch variability and lowers the risk of human error. Moreover, the implementation of solvent recovery systems or a transition to green processing techniques can help mitigate both the economic and environmental burdens associated with conventional organic solvents. Freeze-drying (lyophilization) can be employed to improve the long-term stability of liposomal formulations and significantly reduce logistical costs by minimizing dependence on cold-chain storage. We have reviewed previously the preparation and use of other liposomal antibiotics in antimicrobial therapy [164]. 

Finally, liposomes are prone to lipid oxidation and hydrolysis, leading to potential instability. To enhance their stability and prolong their therapeutic effect, liposomal formulations often require polymeric coating. However, this coating process further increases the complexity of synthesis and production costs, making large-scale application more challenging. In addition, post-processing techniques such as freeze-drying, vacuum freeze-drying, and spray-drying can be applied to the prepared liposomes to enhance their stability.

## Figures and Tables

**Figure 1 antibiotics-14-00728-f001:**
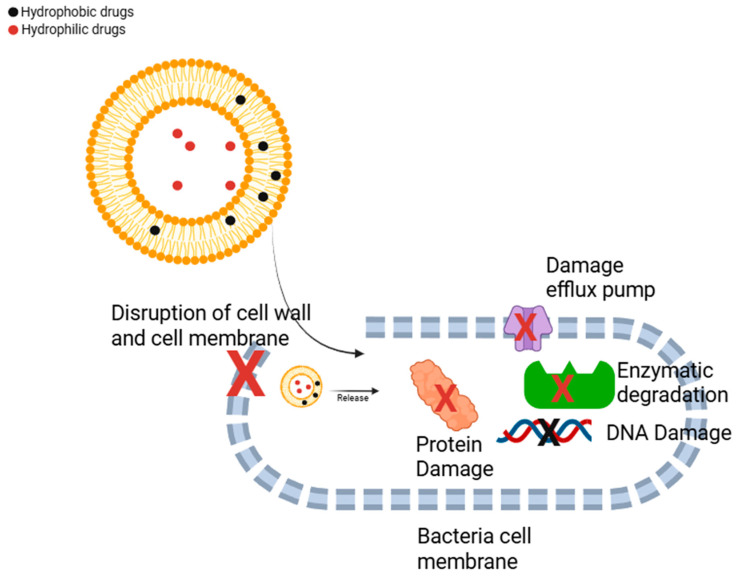
Treatment pathways of antibacterial drug-loaded liposomes against bacteria. (Adapted from [45]). Liposome-encapsulated antibiotics can directly disrupt the bacterial cell wall and cell membrane, leading to loss of structural integrity. These systems are also capable of bypassing efflux pumps, enabling the direct release of the encapsulated drug into the bacterial cytoplasm, which enhances therapeutic efficacy. Moreover, cationic or surface-modified liposomes may contribute to damaging efflux pump structures. Encapsulation within liposomes also protects the antibiotic from enzymatic degradation, thereby increasing its antibacterial activity and stability. By facilitating more efficient intracellular delivery, liposomal systems can enhance the interaction of the antibiotic with bacterial DNA and essential metabolic proteins, ultimately contributing to cell death.

**Figure 2 antibiotics-14-00728-f002:**
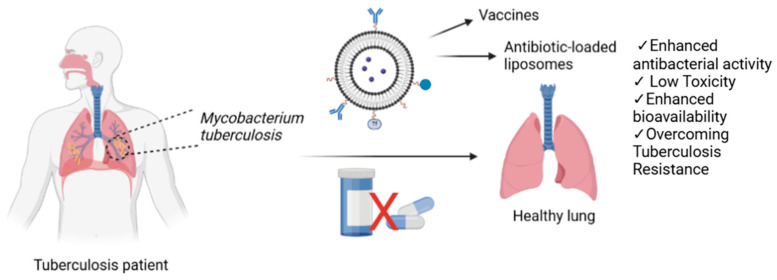
Antibiotic-loaded liposomes as a strategy to overcome tuberculosis resistance and enhance treatment efficacy.

## Data Availability

Not applicable. No new data have been generated for this review.

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
