# Peer review of "Liposome-Encapsulated Antibiotics for the Therapy of Mycobacterial Infections†"

_antibiotics, 2025, doi:10.3390/antibiotics14070728_

Round 1
Reviewer 1 Report
Comments and Suggestions for Authors
1. Abstract needs total revision. The current abstract is more of a brief introduction than a proper structured abstract. It lacks critical components such as background context, objectives, methodology (even for a review), key findings from the literature, and a clear conclusion or implication.
2. Numerous studies are reported regarding liposomal formulations, yet there is limited discussion of comparative efficacy, limitations, or contradictory findings. Please add a comparative table or critical summary (e.g., success rates, delivery methods, challenges) to highlight gaps in the field.
3. The discussion on encapsulation techniques comes across as quite general. It would be helpful if the authors could delve deeper into how specific methods—like thin-film hydration versus supercritical-assisted techniques—affect important factors such as how much drug can be loaded, how the drug is released over time, and how suitable each method is for clinical use. Adding these insights would make the section more informative and relevant.
4. While preclinical findings are well-covered, there is a lack of detail on the current clinical status of liposomal antibiotics for TB (e.g., inhaled amikacin liposome). Include a dedicated section on clinical trials, regulatory status, and translational hurdles (e.g., cost, manufacturing complexity).
5. The discussion related to Table 2 is quite limited. While the table itself includes valuable data on various liposomal strategies against tuberculosis, the manuscript doesn’t really elaborate on the significance of these findings. Highlight key takeaways—such as which formulations show the most promise, what trends are emerging, or how these approaches differ in their mechanisms or clinical potential.
6. It is needed to expand the discussion on liposome-based vaccines for tuberculosis, as detail explanation of antigen presentation, immune response elicitation, and protection efficacy are lack of detail. Authors could expand with mechanistic insights on antigen-liposome interaction, relevant in vivo data, and discuss limitations of current vaccine candidates.
7. Line 259. The term theranostic approaches is mentioned, but it’s not clearly explained or developed in the text. It would be helpful for the authors to briefly describe what this means in the context of tuberculosis treatment and helpful for the readers who may not familiar with the terms.
Author Response
Reviewer 1
Comments 1: Abstract needs total revision. The current abstract is more of a brief introduction than a proper structured abstract. It lacks critical components such as background context, objectives, methodology (even for a review), key findings from the literature, and a clear conclusion or implication.
Response 1: Thank you for this suggestion. We have revised the abstract according to the recommendations of the Reviewer.
Comments 2: Numerous studies are reported regarding liposomal formulations, yet there is limited discussion of comparative efficacy, limitations, or contradictory findings. Please add a comparative table or critical summary (e.g., success rates, delivery methods, challenges) to highlight gaps in the field.
Response 2: We appreciate the concern of the reviewer. Nevertheless, the details of liposomal formulations would be very difficult to extract from the papers. Our main aim in this review is to bring to the attention of the scientific community the progress made in this field at a time when multi-drug-resistant and XDR tuberculosis is becoming a highly significant public health challenge. It appears that the scientific community has jumped on the bandwagon of aerosolized amikacin, although there are many other liposomal antibiotics and additional delivery methods. We wanted to focus on these antibiotics and delivery methods.
Comments 3: The discussion on encapsulation techniques comes across as quite general. It would be helpful if the authors could delve deeper into how specific methods—like thin-film hydration versus supercritical-assisted techniques—affect important factors such as how much drug can be loaded, how the drug is released over time, and how suitable each method is for clinical use. Adding these insights would make the section more informative and relevant.
Response 3: Thank you for your insightful comment regarding the need for a more detailed comparison of encapsulation techniques. In response, we have revised the relevant section of the manuscript to include a more in-depth discussion on how specific methods—particularly thin-film hydration and supercritical fluid-assisted techniques—impact critical parameters such as drug loading capacity, release kinetics, and clinical suitability. These revisions aim to enhance the scientific value and practical relevance of the discussion.
Comments 4: While preclinical findings are well-covered, there is a lack of detail on the current clinical status of liposomal antibiotics for TB (e.g., inhaled amikacin liposome). Include a dedicated section on clinical trials, regulatory status, and translational hurdles (e.g., cost, manufacturing complexity).
Response 4: We thank the reviewer for this valuable suggestion. In accordance with the recommendation, we have added a dedicated section to the manuscript discussing the current clinical status of liposomal antibiotics for tuberculosis, with a particular focus on inhaled amikacin liposomes (Section 7). This new section includes an overview of relevant clinical trials, regulatory approvals, and key translational challenges such as cost-effectiveness, large-scale manufacturing limitations, and delivery system complexity. We believe this addition enhances the clinical relevance and translational depth of the manuscript. We have also described a study on the consistency of aerosolized liposomal amikacin.
Comments 5: The discussion related to Table 2 is quite limited. While the table itself includes valuable data on various liposomal strategies against tuberculosis, the manuscript doesn’t really elaborate on the significance of these findings. Highlight key takeaways—such as which formulations show the most promise, what trends are emerging, or how these approaches differ in their mechanisms or clinical potential.
Response 5: Our intention with Table 2 was to illustrate the variety of approaches utilizing liposomes for the therapy of tuberculosis. In the text, we chose to focus on liposomal antibiotics, while also discussing some vaccine approaches.
Comments 6: It is needed to expand the discussion on liposome-based vaccines for tuberculosis, as detail explanation of antigen presentation, immune response elicitation, and protection efficacy are lack of detail. Authors could expand with mechanistic insights on antigen-liposome interaction, relevant in vivo data, and discuss limitations of current vaccine candidates.
Response 6: In response to the reviewer’s valuable comment, we have expanded the discussion on liposome-based vaccines for tuberculosis by incorporating additional mechanistic insights and relevant data. Specifically, we have elaborated on the processes of antigen presentation and immune response elicitation, highlighting the roles of CD4⁺ and CD8⁺ T-cell activation, dendritic cell maturation, and cytokine induction.
Furthermore, we have integrated in vivo findings demonstrating the immunogenicity and protective efficacy of liposomal vaccines in murine models, including T-cell responses and antibody production. Lastly, a section outlining the current limitations of liposomal vaccine platforms—such as stability issues, limited antigen loading capacity, high production costs, and scale-up challenges—has been added to provide a balanced perspective. We hope these revisions address the reviewer’s concerns and enhance the overall scientific rigor of the manuscript.
Comments 7: Line 259. The term theranostic approaches is mentioned, but it’s not clearly explained or developed in the text. It would be helpful for the authors to briefly describe what this means in the context of tuberculosis treatment and helpful for the readers who may not familiar with the terms.
Response 7: We thank the reviewer for this insightful comment. In response, we have revised the text at the relevant section to include a brief explanation of "theranostic approaches" within the context of tuberculosis treatment. Specifically, we clarified that theranostics refers to the combination of therapeutic and diagnostic functionalities within a single platform—such as liposomes co-loaded with anti-TB drugs and imaging agents—which enables not only the targeted treatment of Mycobacterium tuberculosis infections but also real-time monitoring of drug distribution, disease progression, and potential adverse effects. This integrated strategy offers a promising avenue for personalized and more effective tuberculosis management. We hope this addition enhances clarity for readers who may not be familiar with the term.
Reviewer 2 Report
Comments and Suggestions for Authors
Reviewer Comments to Author
Date: May 09, 2025
Manuscript ID: Antibiotics_3647930
Title: Liposome-Encapsulated Antibiotics for the Therapy of Mycobacterial and Other Bacterial
Infections and Liposomal Vaccines against Tuberculosis
Recommendation: Reject
Dear Authors,
Thank you for submitting your manuscript to Antibiotics. While the topic of liposome-encapsulated antibiotics and vaccines addresses a significant healthcare concern, the current version of your manuscript requires substantial revisions to meet the publication standards of the journal. Below are specific comments for your consideration:
1. Lack of Novelty and Depth
o The manuscript presents a general overview without offering new insights or innovative perspectives. It primarily compiles existing knowledge without critical analysis or synthesis that would advance the field.
2. Justification for Topic Selection
o The rationale for focusing on liposomal antibiotics is unclear, especially given that only
one FDA-approved liposomal antibiotic (Amikacin liposome inhalation suspension) is currently available. Clarifying the unique contribution and relevance of your review is essential.
3. Missing Discussion on Current Developments
o The manuscript lacks a comprehensive discussion of ongoing clinical trials, recent preclinical studies, or pipeline candidates related to liposomal antibiotics. Including such information would enhance the manuscript's relevance and timeliness.
4. Absence of Comparative Analysis
o There is no comparative evaluation of liposomes against other drug delivery systems, such as nanoparticles, dendrimers, or other nanocarriers. A critical assessment of the advantages and limitations of liposomes relative to these systems is recommended.
5. Unconvincing Scope and Conclusions
o While addressing antimicrobial resistance is important, the manuscript's conclusions are speculative and not substantiated by robust data or references to innovative solutions. A more evidence-based approach is necessary.
6. Poor Presentation and Flow
o The manuscript's structure and logical flow are challenging to follow. For instance:
§ The content in lines 236 to 256 should be integrated under the section "4. Encapsulation of antibiotics in liposomes."
§ The section on "Encapsulation of antibiotics in liposomes" should precede topics 2 and 3 to maintain a logical progression.
§ Figure 1 lacks clarity regarding its intended representation. A more detailed illustration is needed to effectively convey the benefits of liposomal antibiotic delivery.
7. Feasibility Challenges Not Addressed Effectively
o While the manuscript acknowledges challenges such as drug leakage, high production costs, and instability, it does not provide supporting data or propose viable solutions. Addressing these issues with evidence-based strategies would strengthen the manuscript.
8. Mechanistic Insights Lacking
o The discussion on antibiotic delivery using liposomes is descriptive and lacks depth regarding the underlying mechanisms. Providing specific explanations of how liposome-based delivery enhances efficacy would add value.
Conclusion:
In its current form, the manuscript does not meet the scientific depth, innovation, and analytical rigor required for publication in Antibiotics. We encourage you to undertake substantial revisions, incorporating the suggestions above, should you choose to resubmit in the future.
Thank you for considering my feedback.
Author Response
Reviewer 2
Recommendation: Reject
Thank you for submitting your manuscript to Antibiotics. While the topic of liposome-encapsulated antibiotics and vaccines addresses a significant healthcare concern, the current version of your manuscript requires substantial revisions to meet the publication standards of the journal. Below are specific comments for your consideration:
Comments 1: Lack of Novelty and Depth
o The manuscript presents a general overview without offering new insights or innovative perspectives. It primarily compiles existing knowledge without critical analysis or synthesis that would advance the field.
Response 1: We sincerely appreciate the insightful comments of the Reviewer, which have contributed significantly to improving the overall quality and depth of the manuscript.
We have added numerous paragraphs on the use of different liposomal antibiotics, and some comments on innovative approaches, such as the blockade of the RES to enhance targeting to the lungs. We felt that it was important for the current generation of active scientists to become aware of the early breakthroughs in the application of liposome-encapsulated antibiotics for the treatment of mycobacterial infections. We believe that the mere awareness of the scientific community of all the work that has been done in this field will lead to additional innovations and translation of the basic research in animal systems to clinical trials. Although we describe in detail clinical studies on aerosolized amikacin, we also wanted to point to additional avenues that are ripe for clinical development.
Comments 2: Justification for Topic Selection
The rationale for focusing on liposomal antibiotics is unclear, especially given that only
one FDA-approved liposomal antibiotic (Amikacin liposome inhalation suspension) is currently available. Clarifying the unique contribution and relevance of your review is essential.
Response 2: The main rationale is the fact that there is a large body of work on the targeted therapy of mycobacterial infections, which was a focus of the laboratory of the senior author, and many other laboratories since the 1970s and 1980s. Although we describe clinical studies on the only FDA approved liposomal antibiotic, it is important for the scientific and medical community to be aware of the large number of studies on the use of numerous antibiotics, and numerous liposomal formulations in the therapy of mycobacterial infections, especially in view of emerging drug-resistant infections that are becoming public health challenges.
Comments 3: Missing Discussion on Current Developments
The manuscript lacks a comprehensive discussion of ongoing clinical trials, recent preclinical studies, or pipeline candidates related to liposomal antibiotics. Including such information would enhance the manuscript's relevance and timeliness.
Response 3: Thank you for your helpful recommendation. We have added “Section 7. Clinical studies on liposomal amikacin.” We have also added a comment regarding the need for the pharmaceutical industry to bring other liposomal antibiotics to the clinic.
Comments 4: Absence of Comparative Analysis
There is no comparative evaluation of liposomes against other drug delivery systems, such as nanoparticles, dendrimers, or other nanocarriers. A critical assessment of the advantages and limitations of liposomes relative to these systems is recommended.
Response 4: We sincerely appreciate your insightful comments, which have contributed significantly to improving the overall quality and depth of the manuscript. In response, we have clarified in the revised manuscript that the examples listed in Table 2 are drawn from recent preclinical studies. We have also added updated information on ongoing clinical trials involving liposomal antibiotics targeting mycobacterial infections, thereby enhancing the relevance and timeliness of the review. Moreover, the section on Future Directions, Challenges, and Limitations have been thoroughly revised and expanded. We believe that expansion of the review to include comparisons with nanoparticles and dendrimers would have made it rather unwieldy. Nevertheless, we will take this suggestion under advisement for a future manuscript.
Comments 5: Unconvincing Scope and Conclusions
While addressing antimicrobial resistance is important, the manuscript's conclusions are speculative and not substantiated by robust data or references to innovative solutions. A more evidence-based approach is necessary.
Response 5: We have revised the manuscript extensively, in the process describing and discussing almost all of the papers in this area. These additions verify the evidence-based approach of the review. We have also introduced references to the potential use of long-chain lipids with favorable thermotropic properties that are derived from marine invertebrates, which we have studied in collaboration with the late Professor Carl Djerassi at Stanford.
Comments 6: Poor Presentation and Flow
The manuscript's structure and logical flow are challenging to follow. For instance:
- 1. The content in lines 236 to 256 should be integrated under the section "4. Encapsulation of antibiotics in liposomes."
- 2.The section on "Encapsulation of antibiotics in liposomes" should precede topics 2 and 3 to maintain a logical progression.
- 3. Figure 1 lacks clarity regarding its intended representation. A more detailed illustration is needed to effectively convey the benefits of liposomal antibiotic delivery.
Response 6: §1. The content referred to by the Reviewer is now in “Section 5. Therapy of Mycobacterium tuberculosis Infection.” §2. Starting the review with encapsulation strategies would have directed the reader’s attention away from the main theme of the review, which is the therapy of mycobacterial infections. We think it is more important to attract the reader’s attention to the exciting early explorations of applying liposome technology to the targeted therapy of M. tuberculosis. §3. We have revised Figure 1 to become more instructive and added more material to the figure legend to be more instructive. Nevertheless, as we have discussed in the revised manuscript, there are many unknowns about the interaction of liposomes with mycobacteria in endocytotic vesicles.
Comments 7: Feasibility Challenges Not Addressed Effectively
While the manuscript acknowledges challenges such as drug leakage, high production costs, and instability, it does not provide supporting data or propose viable solutions. Addressing these issues with evidence-based strategies would strengthen the manuscript.
Response 7: Thank you for your valuable comment. In response, the relevant revisions have been made and the discussed content has been incorporated into the "Future Directions, Challenges, and Limitations" section of the revised manuscript. We believe these additions enhance the clarity and completeness of the manuscript.
Comments 8: Mechanistic Insights Lacking
The discussion on antibiotic delivery using liposomes is descriptive and lacks depth regarding the underlying mechanisms. Providing specific explanations of how liposome-based delivery enhances efficacy would add value.
Response 8: In the sections we have added to the revised manuscript, we have described in detail almost all the studies employing liposomal antibiotics in the therapy of mycobacterial infections. While discussing these studies, we have introduced our current understanding of the mechanisms of the enhanced effects of liposomal antibiotics.
Comments 9: Conclusion:
In its current form, the manuscript does not meet the scientific depth, innovation, and analytical rigor required for publication in Antibiotics. We encourage you to undertake substantial revisions, incorporating the suggestions above, should you choose to resubmit in the future.
Response 9: We sincerely appreciate the detailed comments of the Reviewer, which have helped us to significantly improve the quality of our manuscript. While we understand that the manuscript, in its previous form, may not have met the required standards, we would like to kindly note that all of the concerns raised by all three Reviewers have been carefully addressed in the revised version. We hope that the substantial revisions now reflect the scientific depth, innovation, and analytical rigor expected by Antibiotics.
Reviewer 3 Report
Comments and Suggestions for Authors
Comments and Suggestions for the Authors
The review article “Liposome-Encapsulated Antibiotics for the Therapy of Mycobacterial and Other Bacterial Infections, and Liposomal Vac-3 cines against Tuberculosis” is a fairly good read.
I recommend ‘Major revision’ as the manuscript require serious modifications in the text/figure before publication.
The following are my comments and suggestions to improve the quality of the manuscript,
- Title fairly reflects the main subject of the manuscript. But it needs modification. The title in the current form suggests the meaning that the review is about Liposome-Encapsulated antibiotics for the therapy of Mycobacterial ‘ALONG’ with other bacterial infections, which is not the fact. The review article talks only about Tuberculosis and Non-Tuberculous Mycobacterial Infections. So, authors should remove ‘and Other Bacterial’ portion from the title to make the title more representative of the topic discussed.
- Abstract is good, that summarizes and reflects the work described in the manuscript.
- Key words are adequate.
- The introduction is adequate and comprehensive.
- Page 1, line 29, “(which can remove the stain from many other bacteria)”, did the authors mean acid fast staining ‘distinguish’ the stain from many other bacteria? If yes, authors should rephrase.
- Page 2, line 68, as this is a review article, authors should let the readers know the name of the chromosomal gene (responsible for mycolic acid synthesis) they are trying to mention as the candidate for mutation which causes MDR in M tuberculosis.
- Page 2, line 79-80, check spelling and font (size/italicization) for ‘has been reconized recently as a pathogen’.
- Page 3, line 127, check font.
- Figure 1. Figure legend must be elaborated and re-written. With current legend, the figure doesn’t say anything. There is a serious issue with the figure, which shows release of the antibiotic causes ‘mitochondria damage’; mycobacteria do not have mitochondria’. This wrong fact presented by the authors must be rectified with highest priority, else the manuscript is not suitable for publication.
- Tables are adequate.
- References are adequate and all accounted for.
Author Response
Reviewer 3
The review article “Liposome-Encapsulated Antibiotics for the Therapy of Mycobacterial and Other Bacterial Infections, and Liposomal Vac-3 cines against Tuberculosis” is a fairly good read.
I recommend ‘Major revision’ as the manuscript require serious modifications in the text/figure before publication.
The following are my comments and suggestions to improve the quality of the manuscript,
Comments 1: Title fairly reflects the main subject of the manuscript. But it needs modification. The title in the current form suggests the meaning that the review is about Liposome-Encapsulated antibiotics for the therapy of Mycobacterial ‘ALONG’ with other bacterial infections, which is not the fact. The review article talks only about Tuberculosis and Non-Tuberculous Mycobacterial Infections. So, authors should remove ‘and Other Bacterial’ portion from the title to make the title more representative of the topic discussed.
Response 1: Thank you for your thoughtful and precise comment regarding the manuscript title. In response, we have revised the title to more accurately reflect the scope of the review, which focuses exclusively on Mycobacterium tuberculosis and non-tuberculous mycobacterial (NTM) infections. The phrase “and Other Bacterial” has been removed to eliminate any potential misinterpretation. We believe the updated title now more appropriately represents the content and intent of the manuscript. We appreciate your guidance in improving the clarity and accuracy of our work.
Comments 2: Abstract is good, that summarizes and reflects the work described in the manuscript.
Response 2: We thank the reviewer for the positive feedback on the abstract. We are pleased to know that it effectively summarizes and reflects the content of the manuscript. Your encouraging remarks are greatly appreciated.
Comments 3: Key words are adequate.
Response 3: We thank the Reviewer for the positive comment regarding the keywords. We are glad to hear that they are considered adequate and appropriate for the manuscript.
Comments 4: The introduction is adequate and comprehensive.
Response 4: We greatly appreciate the constructive comments of the Reviewer.
Comments 5: Page 1, line 29, “(which can remove the stain from many other bacteria)”, did the authors mean acid fast staining ‘distinguish’ the stain from many other bacteria? If yes, authors should rephrase.
Response 5: Similar to the decolorization step in the Gram-stain, the decolorization step in the Acid-Fast Staining process employs acid-alcohol to remove the stain from non-mycobacterial genera, leaving the carbol-fuchsin stain on the mycobacteria.
Comments 6: Page 2, line 68, as this is a review article, authors should let the readers know the name of the chromosomal gene (responsible for mycolic acid synthesis) they are trying to mention as the candidate for mutation which causes MDR in M tuberculosis.
Response 6: We thank the Reviewer for this recommendation. In the revised manuscript we have inserted the main genes involved in isoniazid resistance.
Comments 7: Page 2, line 79-80, check spelling and font (size/italicization) for ‘has been reconized recently as a pathogen’.
Response 7: Thank you for this astute observation. We have corrected the italicized letter.
Comments 8: Page 3, line 127, check font.
Response 8: Thank you for this observation. We have corrected the font.
Comments 9: Figure 1. Figure legend must be elaborated and re-written. With current legend, the figure doesn’t say anything. There is a serious issue with the figure, which shows release of the antibiotic causes ‘mitochondria damage’; mycobacteria do not have mitochondria’. This wrong fact presented by the authors must be rectified with highest priority, else the manuscript is not suitable for publication.
Response 9: We apologize for this oversight on our part, and sincerely thank the reviewer for this important observation and constructive feedback. In response, the figure has been completely revised to accurately reflect the biological context, and the misleading implication regarding mitochondrial damage has been corrected. The updated figure legend has also been thoroughly elaborated to ensure clarity and scientific accuracy. We deeply appreciate the reviewer’s careful attention to this critical detail and the opportunity to improve our manuscript accordingly.
Comments 10: Tables are adequate. References are adequate and all accounted for.
Response 10: We greatly appreciate the thorough review that this Reviewer has provided and thank the reviewer for the kind remarks regarding the adequacy of our tables and references. We greatly appreciate your thoughtful evaluation and encouraging feedback.
Round 2
Reviewer 1 Report
Comments and Suggestions for Authors
Thank you for your careful revision. The manuscript now addresses all my previous concerns. I appreciate your efforts and am happy to the revised version of manuscript in its current form.
Author Response
Comment: Thank you for your careful revision. The manuscript now addresses all my previous concerns. I appreciate your efforts and am happy to the revised version of manuscript in its current form.
Response: We appreciate your time in reviewing our manuscript and your approval of the revised version. We believe that the comments of the reviewers have helped to greatly improve the manuscript.
Reviewer 2 Report
Comments and Suggestions for Authors
Reviewer Comments to Author
Date: June 18, 2025
Manuscript ID: Antibiotics_3647930
Title: Liposome-Encapsulated Antibiotics for the Therapy of Mycobacterial and Other Bacterial
Infections, and Liposomal Vaccines against Tuberculosis
Recommendation: Reject
General Observations:
The revised manuscript demonstrates clear improvements. The authors have added historical context, updated clinical trial data, and expanded mechanistic explanations. Presentation aspects such as Figure 1 have been clarified, and the manuscript reflects a sincere attempt to address previous reviewer concerns. The tone is more balanced, and the overall readability has improved.
Critical Comments:
1. The manuscript still lacks sufficient novelty. While comprehensive, it remains largely
descriptive and does not provide new perspectives or critical frameworks that would advance
the field.
2. A comparative discussion of liposomes with other nanocarrier systems (e.g., polymeric
nanoparticles, dendrimers) is missing. A brief comparative analysis would significantly
enhance the value and completeness of the review.
3. The current structure, with historical studies preceding basic concepts such as encapsulation
methods, may reduce clarity for readers unfamiliar with the topic. Reorganizing sections or
improving transitions would improve logical flow.
4. The conclusions remain generalized. The manuscript does not critically assess the limited
clinical translation of liposomal antibiotics beyond Arikayce, which is essential to justify the
optimism conveyed.
Final Remarks:
Despite meaningful revisions, the manuscript does not yet meet the scientific rigor or originality
expected for publication in Antibiotics. The lack of critical analysis, the absence of comparative evaluation, and limited novelty in approach remain significant shortcomings. Therefore, I recommend rejection of the manuscript in its current form.
Author Response
Comments 1: The revised manuscript demonstrates clear improvements. The authors have added historical context, updated clinical trial data, and expanded mechanistic explanations. Presentation aspects such as Figure 1 have been clarified, and the manuscript reflects a sincere attempt to address previous reviewer concerns. The tone is more balanced, and the overall readability has improved.
Response 1: We appreciate the time the reviewer has taken to critically evalaute our manuscript, and their positive comments on the revised version. We have further modified the manuscript in response to the suggestions of the Academic Editor.
Comments 2: The manuscript still lacks sufficient novelty. While comprehensive, it remains largely
descriptive and does not provide new perspectives or critical frameworks that would advance
the field.
Response 2: A search on PubMed with the terms "liposomes, antibiotics, mycobacterium, therapy" produced 213 results, 54 of which were designated as "review." None of these reviews appear to have the scope and detail of our manuscript, and, of course, some of them are quite dated. We appreciate the evaluation that our review is comprehensive. We respectfully disagree with the comment that the review does not provide a new perspective and a critical framework for the field. By showing the successes and limitations of liposomal antibiotics and vaccines, we believe we will lead the reader (possibly in the pharmaceutical industry) to consider the further development of some of the approaches into the clinic.
Comments 3: A comparative discussion of liposomes with other nanocarrier systems (e.g., polymeric
nanoparticles, dendrimers) is missing. A brief comparative analysis would significantly
enhance the value and completeness of the review.
Response 3: In response to the recommendation of the Academic Editor to introduce several references on lipidic nanoparticles, we have introduced a section entitled "lipidic nanoparticles as alternatives to liposomes." While this section does not include a discussion of polymeric nanoparticles and dendrimers, it reminds the reader of alternatives to liposomes. We think that the expansion of the review into discussing these non-lipidic carriers would render the paper unwieldy. Nevertheless, a review on these carriers is an excellent idea and we may take up this suggestion in the near future.
Comments 4: The current structure, with historical studies preceding basic concepts such as encapsulation methods, may reduce clarity for readers unfamiliar with the topic. Reorganizing sections or
improving transitions would improve logical flow.
Response 4: We preferred to introduce the historical developments first, to intrigue the reader to follow up the methods for the generation of liposomes. We believe there is no set or perfect method to write a review, which actually makes the paper more interesting.
Comments 5: The conclusions remain generalized. The manuscript does not critically assess the limited
clinical translation of liposomal antibiotics beyond Arikayce, which is essential to justify the
optimism conveyed.
Response 5: Since we did not introduce a new method of preparing or delivering liposomal antibiotics, our. conclusions were necessarily generalized. Nevertheless, we have added an appeal to the pharmaceutical industry to develop other liposomal antibiotics for refractory lung infections.
Comments 6: Final Remarks: Despite meaningful revisions, the manuscript does not yet meet the scientific rigor or originality expected for publication in Antibiotics. The lack of critical analysis, the absence of comparative evaluation, and limited novelty in approach remain significant shortcomings. Therefore, I recommend rejection of the manuscript in its current form.
Response 6: We respectfully disagree. We believe we have focused on each study we have cited, rather than mentioning them in passing; and thus we have given the due emphasis on each of these papers. We believe a comparative analysis between different carrier systems, different delivery routes, and different antibiotics can be achieved primarily in a study carried out by a particular laboratory using a particular animal model of tuberculosis or non-tuberculous mycobacterial disease. We hope that the reviewer will still have benefitted from our expansive exposé of the field.
Reviewer 3 Report
Comments and Suggestions for Authors
Comments and Suggestions for the Authors
The quality of revised version of the review article “Liposome-Encapsulated Antibiotics for the Therapy of Mycobacterial Infections, and Liposomal Vaccines against Tuberculosis” is now better than the previous version and can be accepted. I am happy that the authors have rectified the figure as well as earlier shortcomings of the manuscript.
Author Response
Comments: The quality of revised version of the review article “Liposome-Encapsulated Antibiotics for the Therapy of Mycobacterial Infections, and Liposomal Vaccines against Tuberculosis” is now better than the previous version and can be accepted. I am happy that the authors have rectified the figure as well as earlier shortcomings of the manuscript.
Response: Thank you for your kind comments on the manuscript. We have now improved it further in response to the suggestions of the Academic Editor.